# "I don´t put people into boxes, but…" A free-listing exercise exploring social categorisation of asylum seekers by professionals in two German reception centres

Sandra Ziegler[1,2]*, Kayvan Bozorgmehr[1,2]

1 Section for Health Equity Studies & Migration, Heidelberg University Hospital, Heidelberg, Germany,
2 Department of Population Medicine and Health Services Research, School of Public Health, University of Bielefeld, Bielefeld, Germany

* Sandra.Ziegler@med.uni-heidelberg.de

**Data Availability Statement:** Relevant data are provided within the paper and its supporting information files. Participants agreed to the processing of their data in a condensed form and

## Abstract

Newly arriving asylum seekers in Germany mostly live in large reception centres, depending on professionals in most aspects of their daily lives. The legal basis for the provision of goods and services allows for discretionary decisions. Given the potential impact of social categorisation on professionals' decisions, and ultimately access to health and social services, we explore the categories used by professionals. We ask of what nature these categorisations are, and weather they align with the public discourse on forced migration. Within an ethnographic study in outpatient clinics of two refugee accommodation centres in Germany, we conducted a modified free-listing with 40 professionals (physicians, nurses, security-personnel, social workers, translators) to explore their categorisation of asylum seekers. Data were qualitatively analysed, and categories were quantitatively mapped using Excel and the Macro "Flame" to show frequencies, ranks, and salience. The four most relevant social categorisations of asylum seekers referred to "demanding and expectant," "polite and friendly" behaviour, "economic refugees," and "integration efforts". In general, sociodemographic variables like gender, age, family status, including countries and regions of origin, were the most significant basis for categorisations (31%), those were often presented combined with other categories. Observations of behaviour and attitudes also influenced categorisations (24%). Professional considerations, e.g., on health, education, adaption or status ranked third (20%). Social categorisation was influenced by public discourses, with evaluations of flight motives, prospects of staying in Germany, and integration potential being thematised in 12% of the categorisations. Professionals therefore might be in danger of being instrumentalised for internal border work. Identifying social categories is important since they structure perception, along their lines deservingness is negotiated, so they potentially influence interaction and decision-making, can trigger empathy and support as well as rejection and discrimination. Larger studies should investigate this further. Free-listing provides a suitable tool for such investigations.

were assured that only example quotes, and short excerpts would be shared after analysis. They did not consent to sharing their complete social categorisation lists with third parties. To adhere to this consent and protect participants from identification, we only share data in the specified format. Compilations of original data are available to eligible researchers upon request: Section for Health Equity Studies & Migration, Heidelberg University Hospital, Im Neuenheimer Feld 130.3, 69120 Heidelberg, SektionEquityMig.AMED@med. uni-heidelberg.de.

**Funding:** This study was funded by the German Federal Ministry of Education and Research (BMBF) in the scope of the RESPOND project (grant no: 01GY1611, grant holder: KB). The funder had no influence on study design, analysis or decision to publish.

**Competing interests:** The authors have declared that no competing interests exist.

## Introduction

Many aspects of the lives of newly arriving refugees in collective accommodation centres are shaped by interactions with and decisions of professionals, such as physicians, nurses, psychologists, social workers, security personnel or interpreters. They act on the basis of expertise, but former experiences also fuel prefabricated patterns of interpretation [1]. Many studies have shown that social categorisation processes based on multiple, consciously or unintentionally applied criteria affect decisions and behaviour of professionals [e.g., 2–10]. Due to time constraints and the high complexity of social spaces–like refugee reception–professionals may need to simplify their social perceptions to be able to act on the grounds of "less-than-perfect information" [11, p. 23]. Social categorisation accomplishes this; by cognitively putting people into clusters according to at least one common characteristic, fading out their individuality and perceiving them as interchangeable members of categories or collectives [12–14] we immediately have an idea of who the other person (presumedly) is, where we are different or alike, what we think of them and how we feel about them [cf. 15]. We tend to overestimate similarities within a category, which simplifies the attribution of common characteristics to the imagined "group." Our thoughts, attitudes, and behaviours are influenced by the judgments we make about this group [16]. This helps to quickly know how to act or react [13, 17–19]. Generalisation increases the likelihood of treating individuals belonging to a specific category in a similar manner [16].

Human differentiations change historically and are dependent on the situation and social field. "Relevant" categories and their meaning are socially constructed [11, 20] they refer to characteristics that a society deems meaningful [21]. Looking at scientific thematisations, "race"/ethnicity and gender seem to be especially meaningful in many societies. A number of categories have been thematised in the health sciences, as impacting on health and healthcare for example: "race"/ethnicity [22–28], gender [29–31], sexual orientation [32–34], social class [35], socio-economic status [e.g., 5], age [e.g., 36, 37], body shape [38, 39] as well as mental state [e.g., 40, 41].

"Immigration" itself has been introduced as social determinant of health [42]. In practice this category gets broken down into ever more internal distinctions, but the (super-)diversity [43, 44] of migrant populations is barely reflected in statistical categories, calling for a more nuanced approach of considering differences by legal status and other characteristics [45]. Societies differentiate "types of migrants [. . .] in relation to each other–as refugees or economic migrants, skilled or unskilled, temporary or permanent, legal or illegal, child or adult" [46, p. 220]. Some studies already address this (progressing) subcategorisations [e.g., 46–50]. Further distinctions of asylum seekers are dynamically negotiated, giving rise to ever more differentiations [e.g., 51–59].

Social categorisation of asylum seekers in Germany occurs within a highly ambivalent discursive space. Like in other countries, refugee reception and therefore also health and social care is permeated by varied, oftentimes conflicting lines of political and societal discourse. National societies and politics show inclusive and exclusive tendencies towards foreigners, welcome-culture, openness, diversity-ideologies and willingness to help clashing with fears over 'national' resources and privileges as well as Othering and xenophobia [60]. Similarly, universal human rights–like the right to health–and international conventions are opposed by securitisation policies and regulatory arguments [61–63]; nation states wanting to attract migrants that seem useful to their economies and keep out those that seems of no use to them [64–66]. In the course of new differentiation practices within this discursive space, new patterns of prejudice, discrimination and inequality are to be expected [43]. Previous studies on common categories of discrimination, such as ethnicity, might not capture emerging categorisations referring to–for example–the economic value or burden of asylum seekers.

## Public discourse and social categorisation

The public discourse is characterised by the mutually reinforcing factors of media–sciences–politics–society [67, p. 188]. It is beyond the scope of this publication to analyse their interaction in detail, all influence the discourse. The interaction between politics, traditional, and online media increasingly shapes the perception of refugees in everyday life [67, p. 199]. Given the significant role of media in framing the discourse on flight migration [68], it is worth examining these frames to then assess if they are reflected in individual social perceptions of asylum seekers. Reporting on flight migration reflects the above-mentioned ambivalence. It is characterised by two frames: one that suggests a threat to security, peace, culture, health, or prosperity [67, 69–72], and another that portrays refugees as suffering victims who have been forced to flee due to circumstances beyond their control [61, p. 1751]. Sympathetic coverage may also highlight benefits to destination communities such as balancing demographic change and providing new labour force. The discourse is characterised by fluctuating cycles. After a brief period in the summer of 2015 when neutral or empathy-inducing reports prevailed, German media returned to the more fear-spreading and problematising coverage [73, 74]. As soon as refugees are statistically present in larger numbers than in previous years, a burden frame gains importance [68, p. 212], suggesting that communities and social systems are unreasonably burdened. A link with economy is established. Within this frame it is assumed that many refugees are not concerned with saving their lives but improving their living standards. Sciences and politics thematise so-called push and pull factors and terms such as "economic migrants," or "asylum tourism" shape the perception of refugees as a financial burden (the latter term emerged in the late 1970s and was picked up by Bavarian politicians in 2014 and 2018) [67, 75]).

Political debates are conveyed to the public through the media [cf. 76, p. 2]. How news is presented can shape the perception of migration as a problem and it can influence public opinion, leading to for example majority acceptance of restrictive migration policies [67, 77]. Conversely, media react to public attitudes [78, p. 324] as political debates respond to societal discourses. Media also mediate the latter.

Multiple representative studies indicate that racist attitudes are widespread in German society. Nearly half of the population (49%) still believes in the existence of human "races", at the same time there is an awareness that it is wrong, to use the term (65%) [79, p. 6]. Another study found that 16.9% of the German population hold negative views towards people based on their skin colour or origin [80]. There is also evidence of racist-capitalist discourses in society: A third of respondents from the first study believed that some ethnic groups are inherently more hardworking than others [79]. Additionally, between 17% [80] and 55% [81] of the population fully or partially agree with the statement that foreigners only come to the country to exploit the welfare state. However, it is important to note that racism is not primarily an individual phenomenon but deeply ingrained in society and its institutions [82].

When a "system of discourses and practices [...] legitimises and reproduces historically developed and current power relations" resulting in unequal treatment or inequality of opportunities that denies certain "groups" access to resources and societal participation while granting privileged access to the excluding group, we can speak of racism [83, 84, p. 29f]. The term encompasses not only biological but also ethnic, religious, or cultural constructions of difference [85]. In times of struggles for hegemonic orders, this form of violence targeting those who have been constructed as Others within the dominant order is increasingly prevalent (see racist violence, Galtung 1998 [86]; cultural violence that makes direct and structural violence appear legitimate or at least not unjust: Galtung 1997 [87, p. 342]. The German scientist Paul Mecheril states: "As long as there is a Europe that secures its borders and pursues migration

and refugee policies that [. . .] let people die, while simultaneously presenting Europe as a place and haven for the preservation of human rights, racism must necessarily exist to legitimise these policies" [88].

Regarding the relationship between societal discourses and social categorisation, it is difficult to determine clear effects of specific actors on others due to the multitude of variables. Indirect effects have been shown, for example negative media coverage reinforcing pre-existing negative attitudes [89]. What can be said for certain, is that media, politics, and sciences influence what is talked about and thought about [cf. 67, p. 198].

And what "the public" thinks about regarding flight migration influences the social categorisations of asylum seekers. It can be assumed that political debates, ambivalent media portrayals of refugees, and culturally and therefore socially available racist, capitalist, hegemonic argumentation patterns are reflected in categorisations since social categorisation draws, among other things, on the relevant discourses present in a society. As described, social categorisation serves a quick and easy cognitive processing of the social environment. Meaning it is influenced by how easy or difficult certain categories and associated attributions are cognitively accessible. This is situation- and context-dependent. In times when there is much discussion about certain types of refugees, it is more likely that people categorise their interaction partners in accordance with these available schemes [90]. Stereotypical perceptions are highly cognitively accessible as they are part of our cultural interpretive frameworks; that is why they seem so "right" and easily influence our judgments and responses towards those we have categorised [16].

## Social categorisation and deservingness

All newly arrived asylum seekers in Germany are entitled to the same (restricted) access to accommodation, food, other goods, and healthcare. However, the vagueness of the legal wording allows street-level bureaucrats to interpret the law. Local administrations, healthcare organisations, social counselling, and welfare offices determine actual access. On this "flip-side-of-rights" [91, p. 96], moral negotiations on who should receive benefits or support take place [92]. This "who" deserves something can refer to individuals or groups–and thus people who have been assigned to a specific social category [93, p. 409].

In deservingness research, different concepts have been developed to determine which variables are involved in the attribution of deservingness [94–98].

1. **Control**: What caused the outcome? What is the reason for the current situation of need? How do I evaluate the action that led to a specific outcome? Is the person/group responsible for their condition or a particular outcome, or do I attribute the outcome to other causes beyond their control? [e.g., 98]

2. **Effort and Reciprocity**: What is the person/group doing to help themselves? What has the person/group done or contributed, or what will they potentially do for me/my collective? Will the person/group do something in return for the help? [97]

3. **Attitude**: How do I evaluate the person/group in question? (Feather p. 5). How compliant, docile/grateful and good-mannered [94, 99], likable, moral do I believe the person/group to be? [92–94].

4. **Identity**: How close or distant do I perceive the person/group to be from the social collectives I belong to? Should this person/group belong to what I consider the ingroup and therefore receive support? [97, 100–103]

5. **Need**: How high/intense is the need, how urgently is help needed? [94, 100]

Nielsen, Morten (2020) [104] found that "deservingness criteria are not detached instruments, but rather part of a sense-making process where individuals construct and classify images of needy groups to justify their judgments about deservingness" [p. 123]. These judgments are always relational and conditional, drawing on various sources of moral insight and experience [91, p. 97]. They can be influenced by individual expectations, attitudes, presumptions, personal and professional values, expertise, beliefs, and experiences. Also so called "common-sense" (a conglomerate of social and cultural imprints) plays a role in deservingness ascriptions. Deservingness assessments are grounded in a "particular social and political context", they may "shift and change in response to new knowledge and evolving circumstances" [ibid.].

As mentioned above, "common sense" also includes socially available ways of distinguishing people. Analyses of deservingness assessments for certain social categories in Europe have shown that immigrants are generally considered least deserving (following the elderly, sick, disabled, and unemployed) [105]. We believe that certain categorisations within the asylum-seeking population may lead to varying deservingness attributions. Assessments of deservingness influence decisions about "what kind of treatment asylum seekers should receive–or whether they should receive treatment at all" [92, p. 2]. Interactions with professionals might be influenced by "implicit assumptions that different social and demographic groups", also within this population, deserve distinct levels, kinds, and qualities of care [ibid.]. Social categorisations, along with the moral and ethical considerations associated with them, can therefore affect access to, quality, and outcomes of care [92, 106].

### Aims and research questions

When social categories of "others" are enriched with evaluations such as better/worse or superior/inferior, assigning value and rank, they become manifestly asymmetric [107, p. 167]. Still, social categorisation does not necessarily imply discrimination [108], it is however a prerequisite for it, as stereotypes and prejudice begin from it [16]. Social categorisation can influence professional attitudes, triggering protective, empowering, and supportive behaviour as well as stigmatisation and discrimination. Since literature on deservingness, vulnerability, or specific protection needs of asylum seekers mostly follows common lines of differentiation, we need to determine which categories are in use in real-world settings of refugee reception, to explore new lines of possible discrimination, that we need to focus on in future studies.

We therefore sought to explore through which lenses asylum seeking patients and clients are seen by the professionals they depend on regarding living conditions and access to healthcare and social counselling. This study set out to answer three questions: 1. *What* social categories do health and other professionals in reception centres for asylum seekers use? 2. What *kind* of categorisations can be found in this context? 3.) How permeable are social categorisations for public discourses, for example on evaluations of reasons for flight, prospects of a positive asylum decision, perceived burdens, or opportunities. Finally, we will discuss how deservingness is negotiated in reference to social categories.

### Materials and methods

To explore social categorisations without pre-defining categories we choose an adapted, verbal free-listing. Free-listing is a "semi-quantitative methodology" [109] developed by cognitive anthropologists to explore semantic or so called 'cultural domains': shared, structured knowledge of a collective regarding a limited section of the world [110] or in other words "[. . .] a set of items, all of which a group of people define as belonging to the same type" [111, p. 116]. The

method has been used to understand the cultural relevancy of certain concepts [for examples see 111–114].

## Study design, sampling, and data collection

Our free-listing exercise was included in a multi-sited ethnographic case study in two refugee-outpatient clinics in two reception facilities of different federal states in Germany: One first reception subordinate to the federal state and one community shelter subordinated to the municipality. Participants were recruited after interviews or informal conversations during field time in 2018/19. They were informed about the role of the researcher, anonymisation of data and the broad aim of the research project. To ensure participants felt safe to speak their minds, statements were not recorded, and no sociodemographic data was gathered in a structured format. An introductory sentence and standardised prompt were given to all participants: "In your daily work live you encounter a lot of patients/clients/inhabitants and handle each case individually, but over time, with experience our mind also creates clusters of people. Therefore, I want to ask you: What kinds of asylum seekers are there?" Except the researchers aim to create a bullet-point-list and the invitation to verbally list everything that comes to mind, no further explanations were given. As supplementary techniques for encouragement, reading back of gathered items and nonspecific prompting were used [115]. There was no limitation regarding list length.

## Ethics statement

Formal verbal or (in case of combination with interviews) written consent was obtained from participants. The ethics committee of the University Hospital (Ethikkommission der Medizinischen Fakultät Heidelberg) approved the study (approval number: S-287/2017)

## Data processing and analysis

The hand-written lists were digitalised in Word and inserted into Excel. Items needed to be unified before quantitative analysis [110] with the Excel Macro "Flame" [116] and Excel. As methodological adaption, we coded full sentences and complex statements preparing them for analysis. (We use the word "coding" for any kind of qualitative categorisation, to avoid confusion with the terminology of "social categorisation"). The results were further explored and structured in multiple steps of qualitative analysis. In detail the process looked as follows:

 Step 1 –Analysis of social categories in Flame

- **Alignment of wording**: We first paraphrased all listed statements in German, and translated it, then the data was condensed it into keywords and keyword combinations (e.g., "cheeky Australians")

- **Pseudonymisation**: Countries and regions of origin were pseudonymised, to avoid recognition and stereotyping. For example, the item "Australians" would be renamed "nation9".

- **Identification of units of meaning**: Keyword combinations were separated into individual units of meaning in their order of mention. If a participant said "there are cheeky people, for example Australians" we would note: *cheeky + plusnation9*. If the nation was the leading categorisation: "there are Australians, many of them are cheeky" it would be *nation9 + cheeky* (for a full, structured item list see S1 File)

- **Analysis 1 in Flame**: The generated items were analysed with Flame v1.2 [116]. For this first frequency analysis, every item is counted once, at first place of occurrence in a list, duplicates are deleted, the frequency measure counts how often a specific item appears in this way over

all lists; the rank indicates the position of an item in each individual list. The items occurring early in the lists of many informants are called "salient" items. Saliency is calculated combining frequency of mention and average rank (avrk.). Of many salience indices, we chose the Sutrop-Index since it is not influenced by the length of lists: SIS (Sutrop-Index-Salience) = F Frequency / (N number of informants x mean rank) [110, 117]. The index can take values between 0 and 1, the latter indicating the highest possible salience (if all participants were to list an item first). Since there is no consensually defined threshold to define which items are salient, we will show results that have a salience value ≥ 0,010 [inspired by 118] when ordering results in this way and frequencies ≥ 10% [119].

- **Analysis 2 in Flame (excluding national or regional specifications)**: In a second analysis step, the specifications of (pseudonymised) nationalities and regions of origin were taken out, only counting if any were mentioned. In German refugee accommodation centres, the composition of inhabitants is based on a central distribution system assigning people of specific nationalities to specific regions and therefore centres, this reduces the significance of the frequency and rank of specific nationality designations.

- **Re-including duplicates to describe actual frequencies of mention**: An additional frequency calculation was done, including duplicates, since Flame deletes those, but we considered mentioning an item more than once as also pointing to relevancy.

- **Analysis 3 in Flame: Comparison of groups (including national and regional specifications)**: To compare results of 1.) health professionals (physicians, nurses, medical assistants), 2.) security and 3.) other staff a further Flame-Analysis was done.

The following three further content analysis steps, were done including duplicates. The aim of this further analysis was to explore the nature of the observed social categorisations, with reference to the original data. All codings were discussed extensively by the authors, to ensure intersubjective plausibility.

Step 2: Exploring themes of categorisations: Further coding and analysis including duplicates

1.) **Inductive coding of themes (super-categories)**: All items were inductively grouped into 25 super categories, to analyse the distribution of broader themes that the categories referred to. If for example someone mentioned there were "polite asylum seekers" this item would get assigned the code "manners and behaviour" (for a "code-book" of this analysis step, see S1 File. The headlines hold the super categories, the table the subcodes or items that were assigned to this code. Items that are not self-explanatory or that summarise certain statements are provided there with quotations. Also, statements that were assigned to an item, but slightly differed in wording from it are provided as quotes.).

2.) **Deductive coding of perspectives (focus-categories):** With the goal to further condense the findings and identify the basic perspective of the categorisation as well as the proportion of items that referred to the public discourse a similar coding process was done again. All original items were newly grouped, now into eight pre-determined focus categories (those were derived from the inductive coding of step 1.): professionally relevant aspects, interactional and behavioural categories, socio-demographical variables, categorisations that referred to the societal/ public/political discourse, adversities and victimhood, appearance, or the self of the categoriser.

A note on the explicable focus category of professional relevancy: For a health professional, the health status, like references to chronic illness, would have counted as a professionally relevant categorisation, whereas to distinguish if people were calm or aggressive would be seen as being more relevant to a security guard; for a government administrative it is important, if

someone understands the system with rights and obligations or not; translators have to consider the educational level of a person, to adopt their language accordingly. If a client comes from a so-called safe country of origin might only be professionally relevant for social workers, since it is their job to manage the legal consequences of this status with their client.

The super- and focus category analysis was also repeated separately for the three groups of professionals mentioned above. Results will be shown in the Supporting Information.

Step 3: Exploring links and combinations of social categorisations

3.) **Qualitative analysis of category-combinations**: In a last step we looked back at the original data, before word alignment, condensation and splitting up of lengthy and complex statements. We went through the original, full-text lists again, especially focusing on complex statements and multiple, successive statements with similar themes, to discern inductively which couplings were presented in the original data. Here we did a standard content analysis, paraphrasing common combinations into summarising sentences. If for example someone said: "There are asylum seekers from nation10 who take care of their kids, but those of nation7 let them run free", we would paraphrase: "Reflection on the protective or neglectful treatment of children of asylum seekers from certain countries of origin." Also, general observations on the full data corpus were noted. The results will be provided in an own chapter of the results section, to ensure the weaknesses of this mainly semi-quantitative approach are balanced out.

## Results

### Participants and data

A total of 40 participants were recruited to free list their social differentiations of asylum seekers. Among them were 14 health professionals, four social workers, five employees of government agencies responsible for social and health benefits, ten security guards, a driver, a caretaker, and a facility manager of a reception facility Table 1.

The 40 lists contained around 470 short or complex statements. Within them 165 *different* social categories or units of social cognition were identified. If nation specifications were not accounted for, a total number of 624 items distributed over the lists was analysed for salience in Flame. (For the descriptives, in case mentions of specific countries or regions are counted separately see S1 Table, column 2). Each list held on average 16 items (min = 4, max = 31 items).

### What social categories do professionals in care settings for asylum seekers use? Most important categorisations according to items frequencies and salience

We present the relevant social categorisations in four tables. As mentioned, firstly national and regional specifications were counted as singular items in Table 2 (ordered by frequencies) and

**Table 1. Participants of the free-listing exercise.**

| profession | | gender | | age | | | | location | |
|---|---|---|---|---|---|---|---|---|---|
| | | m | f | 18–25 | 26–40 | 41–64 | > 65 | loc. 1 | loc. 2 |
| Security | 10 | 9 | 1 | | 4 | 6 | | 6 | 4 |
| Social worker | 4 | 2 | 2 | | 2 | 2 | | 3 | 1 |
| Welfare Adm. | 5 | 1 | 4 | 1 | 2 | 1 | | | 5 |
| Translator | 4 | 3 | 1 | 1 | 3 | | | 3 | 1 |
| Physician | 6 | 4 | 2 | | | 4 | 2 | 5 | 1 |
| Nurse | 8 | | 8 | 1 | 5 | 2 | | 3 | 5 |
| Other | 3 | 1 | 2 | | | 3 | | 1 | 2 |
| Total | 40 | 20 | 20 | 3 | 16 | 18 | 2 | 21 | 19 |

**Table 2. Items ordered by frequency.**

| Original Name | Occurrence Number | Frequency | Summed Ranks | Average Rank | Sutrop Index |
|---|---|---|---|---|---|
| demanding and expectant | 17 | 42.50% | 148 | 8.706 | 0.049 |
| polite and friendly | 15 | 37.50% | 134 | 8.933 | 0.042 |
| integration and/or working effort | 15 | 37.50% | 141 | 9.400 | 0.040 |
| health seeking migrants | 12 | 30.00% | 120 | 10.000 | 0.030 |
| aggressive | 12 | 30.00% | 140 | 11.667 | 0.026 |
| economic refugees | 11 | 27.50% | 78 | 7.091 | 0.039 |
| female | 11 | 27.50% | 97 | 8.818 | 0.031 |
| thankful | 10 | 25.00% | 87 | 8.700 | 0.029 |
| not adapted and insubordinate | 9 | 22.50% | 59 | 6.556 | 0.034 |
| calm | 8 | 20.00% | 55 | 6.875 | 0.029 |
| political refugees | 8 | 20.00% | 59 | 7.375 | 0.027 |
| adapted and subordinate | 8 | 20.00% | 60 | 7.500 | 0.027 |
| war refugees | 8 | 20.00% | 66 | 8.250 | 0.024 |
| plusnation1 | 8 | 20.00% | 81 | 10.125 | 0.020 |
| drug consumers and addicts | 8 | 20.00% | 91 | 11.375 | 0.018 |
| educated | 8 | 20.00% | 95 | 11.875 | 0.017 |
| oppression of woman | 8 | 20.00% | 121 | 15.125 | 0.013 |
| searching for a better life | 7 | 17.50% | 26 | 3.714 | 0.047 |
| refusal to generalise | 7 | 17.50% | 37 | 5.286 | 0.033 |
| youngmale | 7 | 17.50% | 44 | 6.286 | 0.028 |
| needing medical help | 7 | 17.50% | 45 | 6.429 | 0.027 |
| system exploiters | 7 | 17.50% | 59 | 8.429 | 0.021 |
| poor perspective to stay | 7 | 17.50% | 64 | 9.143 | 0.019 |
| male | 7 | 17.50% | 71 | 10.143 | 0.017 |
| no integration and/or working effort | 7 | 17.50% | 77 | 11.000 | 0.016 |
| uneducated | 7 | 17.50% | 82 | 11.714 | 0.015 |
| wanting certificates | 7 | 17.50% | 92 | 13.143 | 0.013 |
| longer there | 7 | 17.50% | 104 | 14.857 | 0.012 |
| plusnation5 | 7 | 17.50% | 109 | 15.571 | 0.011 |
| psychological issues | 7 | 17.50% | 115 | 16.429 | 0.011 |
| nations2 | 6 | 15.00% | 39 | 6.500 | 0.023 |
| scrambling and impatient | 6 | 15.00% | 63 | 10.500 | 0.014 |
| traumatized | 6 | 15.00% | 63 | 10.500 | 0.014 |
| plusnation19 | 6 | 15.00% | 65 | 10.833 | 0.014 |
| plusnations1 | 6 | 15.00% | 80 | 13.333 | 0.011 |
| shorter there | 6 | 15.00% | 82 | 13.667 | 0.011 |
| multiple migrations | 6 | 15.00% | 115 | 19.167 | 0.008 |
| young | 5 | 12.50% | 34 | 6.800 | 0.018 |
| pregnant | 5 | 12.50% | 36 | 7.200 | 0.017 |
| old | 5 | 12.50% | 36 | 7.200 | 0.017 |
| audacious | 5 | 12.50% | 37 | 7.400 | 0.017 |
| equal treatment | 5 | 12.50% | 42 | 8.400 | 0.015 |
| deceiving | 5 | 12.50% | 45 | 9.000 | 0.014 |
| annoying | 5 | 12.50% | 47 | 9.400 | 0.013 |
| gender awareness | 5 | 12.50% | 47 | 9.400 | 0.013 |
| criminals | 5 | 12.50% | 48 | 9.600 | 0.013 |
| plusnation2 | 5 | 12.50% | 51 | 10.200 | 0.012 |

*(Continued)*

**Table 2.** (Continued)

| Original Name | Occurrence Number | Frequency | Summed Ranks | Average Rank | Sutrop Index |
|---|---|---|---|---|---|
| nation19 | 5 | 12.50% | 52 | 10.400 | 0.012 |
| mild diseases/needing general care | 5 | 12.50% | 60 | 12.000 | 0.010 |
| family | 5 | 12.50% | 61 | 12.200 | 0.010 |
| not thankful | 5 | 12.50% | 62 | 12.400 | 0.010 |
| nations4 | 5 | 12.50% | 64 | 12.800 | 0.010 |
| with translator | 5 | 12.50% | 76 | 15.200 | 0.008 |
| plusnation22 | 5 | 12.50% | 87 | 17.400 | 0.007 |
| real refugees needing help | 4 | 10.00% | 21 | 5.250 | 0.019 |
| faking illness | 4 | 10.00% | 26 | 6.500 | 0.015 |
| nation1 | 4 | 10.00% | 27 | 6.750 | 0.015 |
| plusnations2 | 4 | 10.00% | 37 | 9.250 | 0.011 |
| singlefemale | 4 | 10.00% | 39 | 9.750 | 0.010 |
| seriously ill | 4 | 10.00% | 42 | 10.500 | 0.010 |
| nation5 | 4 | 10.00% | 51 | 12.750 | 0.008 |
| confident | 4 | 10.00% | 52 | 13.000 | 0.008 |
| impolite | 4 | 10.00% | 53 | 13.250 | 0.008 |
| fear inducing | 4 | 10.00% | 60 | 15.000 | 0.007 |
| insecure | 4 | 10.00% | 61 | 15.250 | 0.007 |
| chronic disease | 4 | 10.00% | 61 | 15.250 | 0.007 |
| structurally no treatment possible | 4 | 10.00% | 91 | 22.750 | 0.004 |

Table 3 (ordered by salience). For a line graph with both values from this analysis round, see also Fig 1.

Furthermore, Table 4 (frequencies) and 5 (salience) show the results only counting if any reference to the origin of asylum seekers was mentioned (see Methods: Analysis 2 in Flame).

Only the leading social categories (the top 10 of the results) will be reported on in text form here. Four of those items could be considered the most important social categorisations of asylum seekers, since relevance measures overlap according to salience *and* frequency of mention, *and* they made the top 10 regardless if nationalities were specified or not:

1. *Demanding and expectant* asylum seekers: People get assigned to that social category, if they are considered to come before the professionals with high expectations and/or are presenting their concerns or requests in a pressing manner.

2. *Polite and friendly* asylum seekers, are also among the most relevant categories, pointing to people whose social behaviour in the interaction leaves a positive impression.

3. *Integration and/or working effort* seems to also be an important aspect of differentiation; professionals describe observations, like making an effort to learn German, educational level, cleverness, or engagement, and evaluate the "integration potential" or rather adaptive potential of the people in their care.

4. *Economic refugees*: People get assigned that category if the categoriser assumes, they fled poverty or a lack of opportunities–in contrast to war or political reasons for fleeing (see S1 File for quotes from the data regarding this item).

As soon as national or regional specifications were taken out of the calculations, just counting if any country or region was mentioned, also *nationalities* either as main descriptor of

**Table 3. Items ordered by salience.**

| Original Name | Occurrence Number | Frequency | Summed Ranks | Average rank | Sutrop Index |
|---|---|---|---|---|---|
| demanding and expectant | 17 | 42.50% | 148 | 8.706 | 0.049 |
| searching for a better life | 7 | 17.50% | 26 | 3.714 | 0.047 |
| polite and friendly | 15 | 37.50% | 134 | 8.933 | 0.042 |
| integration and/or working effort | 15 | 37.50% | 141 | 9.400 | 0.040 |
| economic refugees | 11 | 27.50% | 78 | 7.091 | 0.039 |
| not adapted and insubordinate | 9 | 22.50% | 59 | 6.556 | 0.034 |
| refusal to generalize | 7 | 17.50% | 37 | 5.286 | 0.033 |
| female | 11 | 27.50% | 97 | 8.818 | 0.031 |
| health seeking migrants | 12 | 30.00% | 120 | 10.000 | 0.030 |
| calm | 8 | 20.00% | 55 | 6.875 | 0.029 |
| thankful | 10 | 25.00% | 87 | 8.700 | 0.029 |
| youngmale | 7 | 17.50% | 44 | 6.286 | 0.028 |
| needing medical help | 7 | 17.50% | 45 | 6.429 | 0.027 |
| political refugees | 8 | 20.00% | 59 | 7.375 | 0.027 |
| adapted and subordinate | 8 | 20.00% | 60 | 7.500 | 0.027 |
| aggressive | 12 | 30.00% | 140 | 11.667 | 0.026 |
| nation10 | 1 | 2.50% | 1 | 1.000 | 0.025 |
| seeking asylum advice | 1 | 2.50% | 1 | 1.000 | 0.025 |
| nation18 | 1 | 2.50% | 1 | 1.000 | 0.025 |
| victims of violence | 1 | 2.50% | 1 | 1.000 | 0.025 |
| war refugees | 8 | 20.00% | 66 | 8.250 | 0.024 |
| nations2 | 6 | 15.00% | 39 | 6.500 | 0.023 |
| system exploiters | 7 | 17.50% | 59 | 8.429 | 0.021 |
| plusnation1 | 8 | 20.00% | 81 | 10.125 | 0.020 |
| poor perspective to stay | 7 | 17.50% | 64 | 9.143 | 0.019 |
| real refugees needing help | 4 | 10.00% | 21 | 5.250 | 0.019 |
| young | 5 | 12.50% | 34 | 6.800 | 0.018 |
| drug consumers and addicts | 8 | 20.00% | 91 | 11.375 | 0.018 |
| pregnant | 5 | 12.50% | 36 | 7.200 | 0.017 |
| old | 5 | 12.50% | 36 | 7.200 | 0.017 |
| male | 7 | 17.50% | 71 | 10.143 | 0.017 |
| audacious | 5 | 12.50% | 37 | 7.400 | 0.017 |
| educated | 8 | 20.00% | 95 | 11.875 | 0.017 |
| no integration and/or working effort | 7 | 17.50% | 77 | 11.000 | 0.016 |
| faking illness | 4 | 10.00% | 26 | 6.500 | 0.015 |
| nationality | 3 | 7.50% | 15 | 5.000 | 0.015 |
| uneducated | 7 | 17.50% | 82 | 11.714 | 0.015 |
| equal treatment | 5 | 12.50% | 42 | 8.400 | 0.015 |
| nation1 | 4 | 10.00% | 27 | 6.750 | 0.015 |
| scrambling and impatient | 6 | 15.00% | 63 | 10.500 | 0.014 |
| traumatized | 6 | 15.00% | 63 | 10.500 | 0.014 |
| deceiving | 5 | 12.50% | 45 | 9.000 | 0.014 |
| plusnation19 | 6 | 15.00% | 65 | 10.833 | 0.014 |
| wanting certificates | 7 | 17.50% | 92 | 13.143 | 0.013 |
| annoying | 5 | 12.50% | 47 | 9.400 | 0.013 |
| gender awareness | 5 | 12.50% | 47 | 9.400 | 0.013 |
| often met and/or familiar | 3 | 7.50% | 17 | 5.667 | 0.013 |

*(Continued)*

**Table 3.** (Continued)

| Original Name | Occurrence Number | Frequency | Summed Ranks | Average rank | Sutrop Index |
|---|---|---|---|---|---|
| victims of sexual violence | 3 | 7.50% | 17 | 5.667 | 0.013 |
| oppression of woman | 8 | 20.00% | 121 | 15.125 | 0.013 |
| criminals | 5 | 12.50% | 48 | 9.600 | 0.013 |
| gift-giving | 2 | 5.00% | 8 | 4.000 | 0.013 |
| different disciplines | 1 | 2.50% | 2 | 2.000 | 0.013 |
| plusnation20 | 1 | 2.50% | 2 | 2.000 | 0.013 |
| plusnation2 | 5 | 12.50% | 51 | 10.200 | 0.012 |
| nation19 | 5 | 12.50% | 52 | 10.400 | 0.012 |
| longer there | 7 | 17.50% | 104 | 14.857 | 0.012 |
| plusnations1 | 6 | 15.00% | 80 | 13.333 | 0.011 |
| nations3 | 3 | 7.50% | 20 | 6.667 | 0.011 |
| plusnation5 | 7 | 17.50% | 109 | 15.571 | 0.011 |
| nation15 | 2 | 5.00% | 9 | 4.500 | 0.011 |
| shorter there | 6 | 15.00% | 82 | 13.667 | 0.011 |
| plusnations2 | 4 | 10.00% | 37 | 9.250 | 0.011 |
| psychological issues | 7 | 17.50% | 115 | 16.429 | 0.011 |
| mild diseases/needing general care | 5 | 12.50% | 60 | 12.000 | 0.010 |
| singlefemale | 4 | 10.00% | 39 | 9.750 | 0.010 |
| family | 5 | 12.50% | 61 | 12.200 | 0.010 |
| not thankful | 5 | 12.50% | 62 | 12.400 | 0.010 |
| vulnerable and in-need-of-protection | 2 | 5.00% | 10 | 5.000 | 0.010 |
| good perspective to stay | 2 | 5.00% | 10 | 5.000 | 0.010 |
| jealous | 2 | 5.00% | 10 | 5.000 | 0.010 |
| nations4 | 5 | 12.50% | 64 | 12.800 | 0.010 |
| seriously ill | 4 | 10.00% | 42 | 10.500 | 0.010 |

people or combined with other variables were among the highest ranked social categories according to frequencies *and* salience-index, pushing other items from the top 10 list.

Further social categorisations, that appeared in most analytical steps among the top 10 were *health seeking migrants* (among the top 10 in Tables 2–4), referring to people seeking existing, affordable, or better healthcare than in countries of origin. As an asylum seeker, you furthermore seem to be classified by many professionals according to whether you are able to *adapt and subordinate*, especially focussing on perceived resistance (Tables 2, 3 and 5). Being *female* (Tables 2–4) also seems to be a highly relevant descriptor. Two items of the highest ranked could be regarded as outliers, since frequency and salience measures did not align in the above mentioned way, but they still appear in two of four tables among these high ranked items: 12 of 40 participants mentioned *aggressive* asylum seekers as a category (Tables 2 and 4), but later in their lists (mean rank 11.7) that is why it does not appear among the categories with highest cognitive salience. The other way around, the explanation that there were asylum seekers who migrated, because they were *searching for a better life*–which was assessed neutrally or with empathic understanding–was only mentioned by seven participants, but early in their lists (mean rank 3.7), with the effect that this item appeared among the high ranks, only if the results were ordered by salience.

With a similar effect, seven out of 40 participants explicitly stated that they would *try to avoid putting people into boxes* (in German: "Schubladen" = "drawers"). Additionally, five people assured the interviewer, that they would *treat everyone the same*. These statements are not

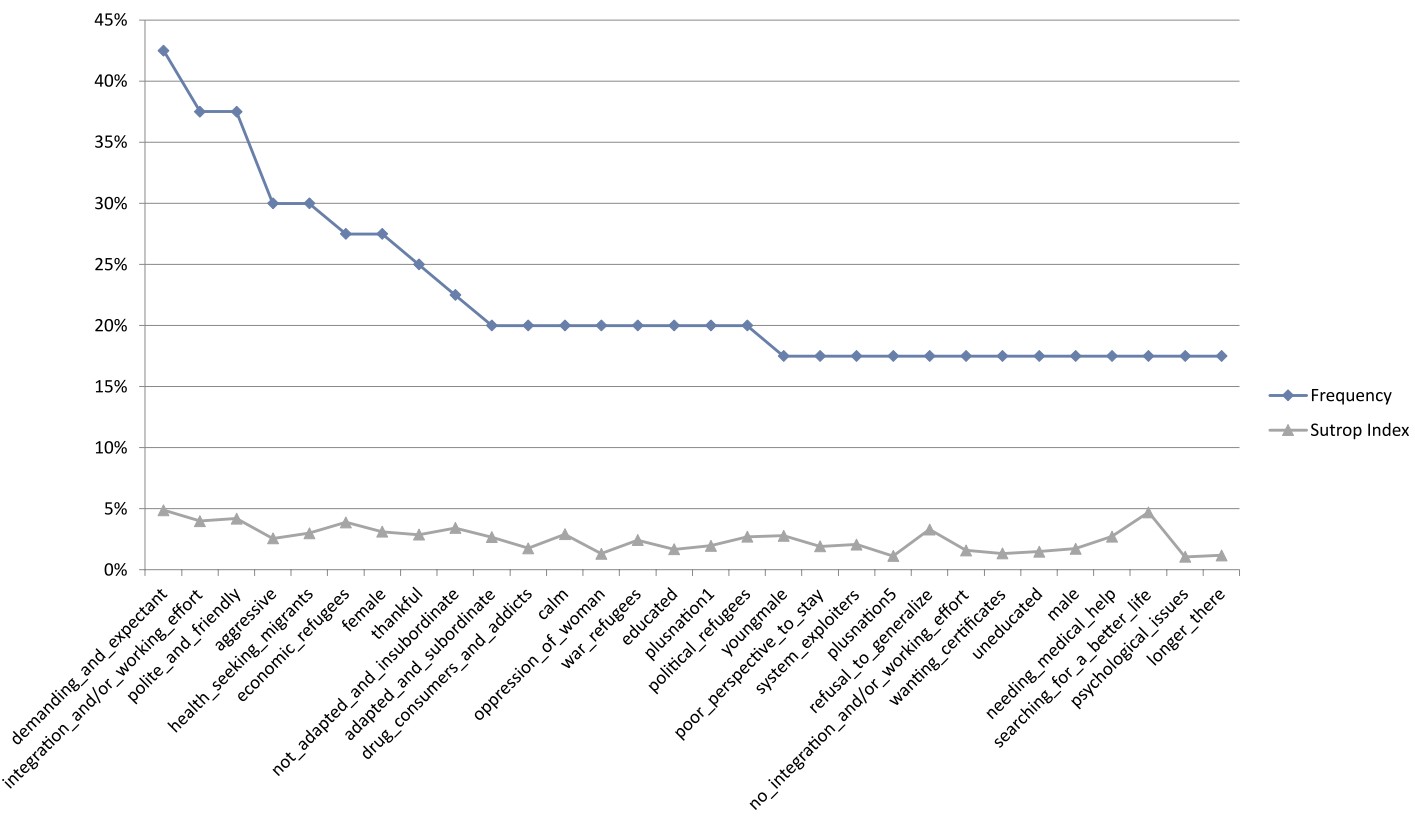

**Fig 1. Items´ frequencies and salience (including nation-specifications).**

considered social categorisation, but an attempt to position oneself while explicating own categorisations, therefore they were included in the analysis. This discomfort of pigeonholing people appeared in the top-10 according to salience (Tables 3 and 5).

If we include duplicates again and count the total frequencies of mention, the general trends within top 10 lists stay stable (S2 Table). Again, we see, that asylum seekers are predominantly categorised according to their demanding or friendly behaviour as well as nationalities or regions of origin, oftentimes appearing in combination.

## What social categories were mentioned most frequently by health care professionals, compared to security and other professionals?

Health professionals most frequently referred to "health seeking migration" (8 of 14 lists held this item, either in evaluative or not evaluative form, see qualitative results). Six of 14 health professionals furthermore mentioned, there were "traumatised" asylum seekers (SIS 0.041) and those who have "psychological issues" (SIS 0.026), as well as "demanding and expectant" asylum seekers (SIS 0.040), also those who would "want certificates" (SIS 0.032) were mentioned with the same frequency. Security personnel relatively often (4 of 10 cases) referred to adaption or subordination, calm or aggressive behaviour as well as "economic refugees" and one specific nation. In the lists of the other staff, like social workers, administrators, or translators 10 of 26 people mentioned "polite and friendly" asylum seekers. "Demanding and expectant" as well as "female" asylum seekers were also common categories here, together with categorisations referring to integration and/or working effort (all with 8 of 26 lists mentioning

**Table 4. Items ordered by frequency (excluding national/regional specifications).**

| Original Name | Occurrence Number | Frequency | Summed Ranks | Average rank | Sutrop Index |
|---|---|---|---|---|---|
| plusnation | 21 | 52.50% | 176 | 8.381 | 0.063 |
| demanding and expectant | 17 | 42.50% | 135 | 7.941 | 0.054 |
| nation | 15 | 37.50% | 105 | 7.000 | 0.054 |
| polite and friendly | 15 | 37.50% | 125 | 8.333 | 0.045 |
| integration and/or working effort | 15 | 37.50% | 128 | 8.533 | 0.044 |
| nations | 12 | 30.00% | 89 | 7.417 | 0.040 |
| health seeking migrants | 12 | 30.00% | 110 | 9.167 | 0.033 |
| aggressive | 12 | 30.00% | 125 | 10.417 | 0.029 |
| economic refugees | 11 | 27.50% | 67 | 6.091 | 0.045 |
| female | 11 | 27.50% | 91 | 8.273 | 0.033 |
| plusnations | 11 | 27.50% | 118 | 10.727 | 0.026 |
| thankful | 10 | 25.00% | 75 | 7.500 | 0.033 |
| not adapted and insubordinate | 9 | 22.50% | 51 | 5.667 | 0.040 |
| adapted and subordinate | 8 | 20.00% | 53 | 6.625 | 0.030 |
| calm | 8 | 20.00% | 54 | 6.750 | 0.030 |
| political refugees | 8 | 20.00% | 58 | 7.250 | 0.028 |
| drug consumers and addicts | 8 | 20.00% | 81 | 10.125 | 0.020 |
| educated | 8 | 20.00% | 87 | 10.875 | 0.018 |
| oppression of woman | 8 | 20.00% | 105 | 13.125 | 0.015 |
| searching for a better life | 7 | 17.50% | 24 | 3.429 | 0.051 |
| refusal to generalise | 7 | 17.50% | 36 | 5.143 | 0.034 |
| youngmale | 7 | 17.50% | 39 | 5.571 | 0.031 |
| needing medical help | 7 | 17.50% | 45 | 6.429 | 0.027 |
| war refugees | 7 | 17.50% | 53 | 7.571 | 0.023 |
| system exploiters | 7 | 17.50% | 56 | 8.000 | 0.022 |
| poor perspective to stay | 7 | 17.50% | 57 | 8.143 | 0.021 |
| male | 7 | 17.50% | 66 | 9.429 | 0.019 |
| no integration and/or working effort | 7 | 17.50% | 69 | 9.857 | 0.018 |
| uneducated | 7 | 17.50% | 74 | 10.571 | 0.017 |
| wanting certificates | 7 | 17.50% | 82 | 11.714 | 0.015 |
| longer there | 7 | 17.50% | 88 | 12.571 | 0.014 |
| psychological issues | 7 | 17.50% | 98 | 14.000 | 0.013 |
| traumatized | 6 | 15.00% | 51 | 8.500 | 0.018 |
| scrambling and impatient | 6 | 15.00% | 56 | 9.333 | 0.016 |
| shorter there | 6 | 15.00% | 68 | 11.333 | 0.013 |
| multiple migrations | 6 | 15.00% | 98 | 16.333 | 0.009 |
| equal treatment | 5 | 12.50% | 31 | 6.200 | 0.020 |
| young | 5 | 12.50% | 31 | 6.200 | 0.020 |
| pregnant | 5 | 12.50% | 34 | 6.800 | 0.018 |
| old | 5 | 12.50% | 36 | 7.200 | 0.017 |
| annoying | 5 | 12.50% | 36 | 7.200 | 0.017 |
| deceiving | 5 | 12.50% | 37 | 7.400 | 0.017 |
| audacious | 5 | 12.50% | 37 | 7.400 | 0.017 |
| criminals | 5 | 12.50% | 39 | 7.800 | 0.016 |
| gender awareness | 5 | 12.50% | 47 | 9.400 | 0.013 |
| mild diseases/needing general care | 5 | 12.50% | 55 | 11.000 | 0.011 |
| not thankful | 5 | 12.50% | 58 | 11.600 | 0.011 |

*(Continued)*

**Table 4.** (*Continued*)

| Original Name | Occurrence Number | Frequency | Summed Ranks | Average rank | Sutrop Index |
|---|---|---|---|---|---|
| family | 5 | 12.50% | 61 | 12.200 | 0.010 |
| with translator | 5 | 12.50% | 65 | 13.000 | 0.010 |
| real refugees needing help | 4 | 10.00% | 18 | 4.500 | 0.022 |
| faking illness | 4 | 10.00% | 26 | 6.500 | 0.015 |
| singlefemale | 4 | 10.00% | 36 | 9.000 | 0.011 |
| seriously ill | 4 | 10.00% | 38 | 9.500 | 0.011 |
| fear inducing | 4 | 10.00% | 48 | 12.000 | 0.008 |
| confident | 4 | 10.00% | 50 | 12.500 | 0.008 |
| chronic disease | 4 | 10.00% | 52 | 13.000 | 0.008 |
| impolite | 4 | 10.00% | 53 | 13.250 | 0.008 |
| insecure | 4 | 10.00% | 54 | 13.500 | 0.007 |
| structurally no treatment possible | 4 | 10.00% | 79 | 19.750 | 0.005 |

it). (For tables with the top-10 items according to frequency of health professionals, security and other staff see S2 File).

## Can evaluations of the public discourse regarding flight motives and prospects of staying be identified in social categorisations of professionals?

Looking again on our full sample, we could identify categorisations of asylum seekers according to suspected flight motives. In the salience analysis (Table 3), *searching for a better life* was among the most salient items (0.047), 18% of all lists contained this motive, on average on the fourth position. Explicitly stating the term *"economic refugees"* were 27% of the participants, this category always emerged among the ten most relevant social categories, regardless of our mode of analysis, the salience was therefore also high (0.039), it emerged on average in the seventh position of individual lists. Even more professionals (30%) referred to a form of *health seeking migration* (SIS 0.030) but on average later in their lists (average rank 10). 20% of the participants referred to *political refugees* as a category (SIS 0.027, average rank 8), *war refugees* were mentioned by 20% (SIS 0.024, avrk 8). One professional also mentioned *religious refugees* (SIS 0.002), others referred to victimhood of *discrimination* (SIS 0.003), *torture* (SIS 0.005) or *violence* (SIS 0.025) as flight motivations.

Related to such social discourses, we also noted depictions of *real refugees needing help* (10%, SIS 0.019, avrk. 5) and the insinuation asylum seekers could be there to *exploit the system* (18%, SIS 0.021, avrk 8). *Poor prospects of staying* were mentioned by 18% of the participants (SIS 0.019, avrk 9) whereas *good prospects* only by 10% (SIS 0.010, avrk 5), two professionals explicitly referred to asylum seekers coming from so called *"safe countries of origin"* (5%, SIS 0.003). Less frequently categorisations also mirrored discourses about gender relations, cultural distance, as well as security considerations (albeit with focus on individual or organisational security, see qualitative results below).

## What kind of categorisations can be found in this context? Analysis of broader themes (super-categories)

The analysis of super-categories (see headlines S1 File) (Fig 2) shows, that asylum seekers were most commonly categorised according to 1) attitudes attributed to them by professionals, for example whether they were perceived as grateful or demanding, patient or not, insubordinate

**Table 5. Items ordered by salience (excluding national/regional specifications).**

| Original Name | Occurrence Number | Frequency | Summed Ranks | Average rank | Sutrop Index |
|---|---|---|---|---|---|
| plusnation | 21 | 52.50% | 176 | 8.381 | 0.063 |
| nation | 15 | 37.50% | 105 | 7.000 | 0.054 |
| demanding and expectant | 17 | 42.50% | 135 | 7.941 | 0.054 |
| searching for a better life | 7 | 17.50% | 24 | 3.429 | 0.051 |
| economic refugees | 11 | 27.50% | 67 | 6.091 | 0.045 |
| polite and friendly | 15 | 37.50% | 125 | 8.333 | 0.045 |
| integration and/or working effort | 15 | 37.50% | 128 | 8.533 | 0.044 |
| nations | 12 | 30.00% | 89 | 7.417 | 0.040 |
| not adapted and insubordinate | 9 | 22.50% | 51 | 5.667 | 0.040 |
| refusal to generalise | 7 | 17.50% | 36 | 5.143 | 0.034 |
| thankful | 10 | 25.00% | 75 | 7.500 | 0.033 |
| female | 11 | 27.50% | 91 | 8.273 | 0.033 |
| health seeking migrants | 12 | 30.00% | 110 | 9.167 | 0.033 |
| youngmale | 7 | 17.50% | 39 | 5.571 | 0.031 |
| adapted and subordinate | 8 | 20.00% | 53 | 6.625 | 0.030 |
| calm | 8 | 20.00% | 54 | 6.750 | 0.030 |
| aggressive | 12 | 30.00% | 125 | 10.417 | 0.029 |
| political refugees | 8 | 20.00% | 58 | 7.250 | 0.028 |
| needing medical help | 7 | 17.50% | 45 | 6.429 | 0.027 |
| plusnations | 11 | 27.50% | 118 | 10.727 | 0.026 |
| seeking asylum advice | 1 | 2.50% | 1 | 1.000 | 0.025 |
| victims of violence | 1 | 2.50% | 1 | 1.000 | 0.025 |
| war refugees | 7 | 17.50% | 53 | 7.571 | 0.023 |
| real refugees needing help | 4 | 10.00% | 18 | 4.500 | 0.022 |
| system exploiters | 7 | 17.50% | 56 | 8.000 | 0.022 |
| poor perspective to stay | 7 | 17.50% | 57 | 8.143 | 0.021 |
| equal treatment | 5 | 12.50% | 31 | 6.200 | 0.020 |
| young | 5 | 12.50% | 31 | 6.200 | 0.020 |
| drug consumers and addicts | 8 | 20.00% | 81 | 10.125 | 0.020 |
| male | 7 | 17.50% | 66 | 9.429 | 0.019 |
| educated | 8 | 20.00% | 87 | 10.875 | 0.018 |
| pregnant | 5 | 12.50% | 34 | 6.800 | 0.018 |
| no integration and/or working effort | 7 | 17.50% | 69 | 9.857 | 0.018 |
| traumatized | 6 | 15.00% | 51 | 8.500 | 0.018 |
| old | 5 | 12.50% | 36 | 7.200 | 0.017 |
| annoying | 5 | 12.50% | 36 | 7.200 | 0.017 |
| deceiving | 5 | 12.50% | 37 | 7.400 | 0.017 |
| audacious | 5 | 12.50% | 37 | 7.400 | 0.017 |
| uneducated | 7 | 17.50% | 74 | 10.571 | 0.017 |
| scrambling and impatient | 6 | 15.00% | 56 | 9.333 | 0.016 |
| criminals | 5 | 12.50% | 39 | 7.800 | 0.016 |
| faking illness | 4 | 10.00% | 26 | 6.500 | 0.015 |
| oppression of woman | 8 | 20.00% | 105 | 13.125 | 0.015 |
| nationality | 3 | 7.50% | 15 | 5.000 | 0.015 |
| wanting certificates | 7 | 17.50% | 82 | 11.714 | 0.015 |
| longer there | 7 | 17.50% | 88 | 12.571 | 0.014 |
| gender awareness | 5 | 12.50% | 47 | 9.400 | 0.013 |

*(Continued)*

**Table 5.** (Continued)

| Original Name | Occurrence Number | Frequency | Summed Ranks | Average rank | Sutrop Index |
|---|---|---|---|---|---|
| shorter there | 6 | 15.00% | 68 | 11.333 | 0.013 |
| often met and/or familiar | 3 | 7.50% | 17 | 5.667 | 0.013 |
| victims of sexual violence | 3 | 7.50% | 17 | 5.667 | 0.013 |
| psychological issues | 7 | 17.50% | 98 | 14.000 | 0.013 |
| gift-giving | 2 | 5.00% | 8 | 4.000 | 0.013 |
| different disciplines | 1 | 2.50% | 2 | 2.000 | 0.013 |
| mild diseases/needing general care | 5 | 12.50% | 55 | 11.000 | 0.011 |
| singlefemale | 4 | 10.00% | 36 | 9.000 | 0.011 |
| vulnerable/in-need-of-protection | 2 | 5.00% | 9 | 4.500 | 0.011 |
| not thankful | 5 | 12.50% | 58 | 11.600 | 0.011 |
| seriously ill | 4 | 10.00% | 38 | 9.500 | 0.011 |
| family | 5 | 12.50% | 61 | 12.200 | 0.010 |
| alone | 3 | 7.50% | 22 | 7.333 | 0.010 |
| good perspective to stay | 2 | 5.00% | 10 | 5.000 | 0.010 |
| jealous | 2 | 5.00% | 10 | 5.000 | 0.010 |
| with translator | 5 | 12.50% | 65 | 13.000 | 0.010 |

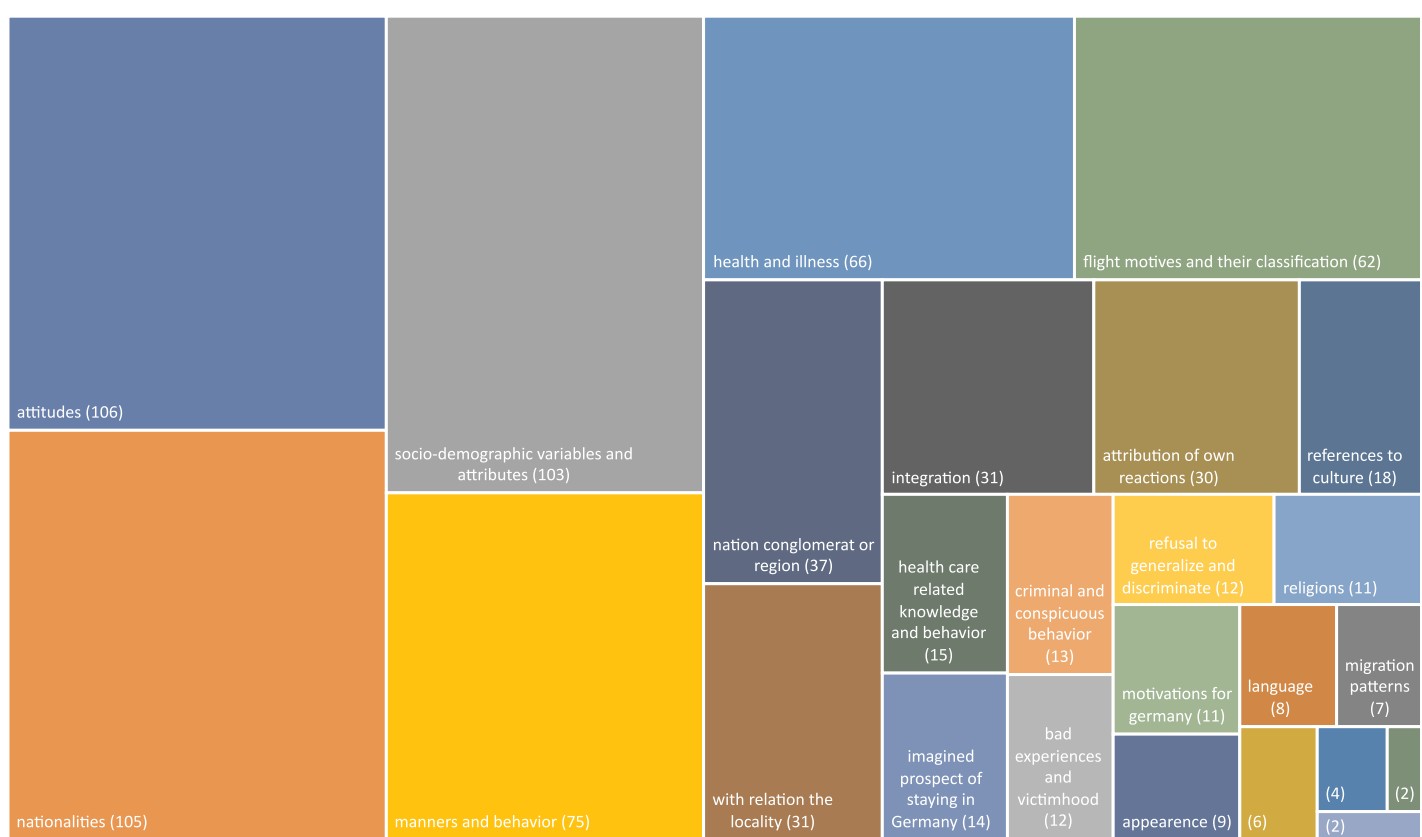

**Fig 2. Analysis of super-categories (including duplicates).**

or respectful of rules, socially distanced or open. 2) general socio-demographic variables fed into categorisation processes, like age, gender, relationship or family status and educational level. Also, 3.) the national origin seems to play an important role for social categorisation; followed by 4.) observed manners and behaviour, like being friendly and polite or not, calm, or loud, and imagined as honest or deceiving. On fifth position–probably due to the fact, that the field study was based within outpatient-clinics–are categorisations of (5.) the condition, on the continuum of health and illness, e.g., having physical or psychological issues, evaluations of the severity of a condition, like having an acute, chronic, or terminal condition. On sixth position we find social differentiations according to different (6.) flight motives: live chances, economic, political, war induced, health related (see above) also the need for help or special protection is summarised in this super-category.

Additional to the mentioned nationalities of origin, also 7.) broader geographical areas like "the Balkans", "Arabia", "Africa" or "MAHGREB" were frequently named by professionals in their categorisations of asylum seekers. Furthermore, 8.) mutual familiarity in relation to the locality seems to also play a role in categorisations: is someone longer there or newly arrived? Have I met them repeatedly? How familiar are they with the facility and its organisations? Do they understand local systems and rules or not?

We also find categorisations according to 9.) observed integration efforts or the suspected integration potential. And 10.) own inner reactions towards asylum seekers can also be attributed to them as categories, for example if professionals feel annoyed, exhausted, compassionate, burdened, anxious or feel they can rely on the other person, they describe annoying and exhausting asylum seekers, ones that trigger compassion and dangerous or trustworthy people.

### Analysis of underlying perspectives (focus-categories)

In our attempt to assess how permeable social categorisations are for the societal, public, and political discourse (see Fig 3) we found that 12% (of the items of obtained lists, inclusive duplicates) related to such discourses concerned with evaluations of flight motives, prospects of staying and the presumed "integration" potential. We also found out which proportion of the categorisation process is fed by observations that are of professional relevancy for the categoriser: 20% of the mentioned items referred to such categories (for a short explanation for professional relevancy, see methods).

Conclusive with the findings from the super-category analysis, the most relevant basis for categorisations, seem to be socio-demographic variables (31%) (in this analysis step containing also countries of origin) and observations of the behaviour in interactions and attitudes derived from it (24%). We also can show, that 6% of the categorisations of others relate to the self of the categoriser, perceived victimhood of asylum seekers (3%) or their external appearance (1%). In 2% of the obtained items forms of discomfort were uttered regarding putting people into "boxes", where professionals stated they would refuse to generalise, treat everyone equally and problematised wrong media images of asylum seekers. (For a comparison of three groups of professionals see S2 File).

### Qualitative analysis of dualities and combinations of categorisations

In an additional qualitative analysis of the data, two observations stand out. Firstly, categorisations repeatedly come in antagonistic pairs, where oppositely attributed characteristics are juxtaposed. Either by naming opposing categories directly one after the other (/) or by combining them into one statement.

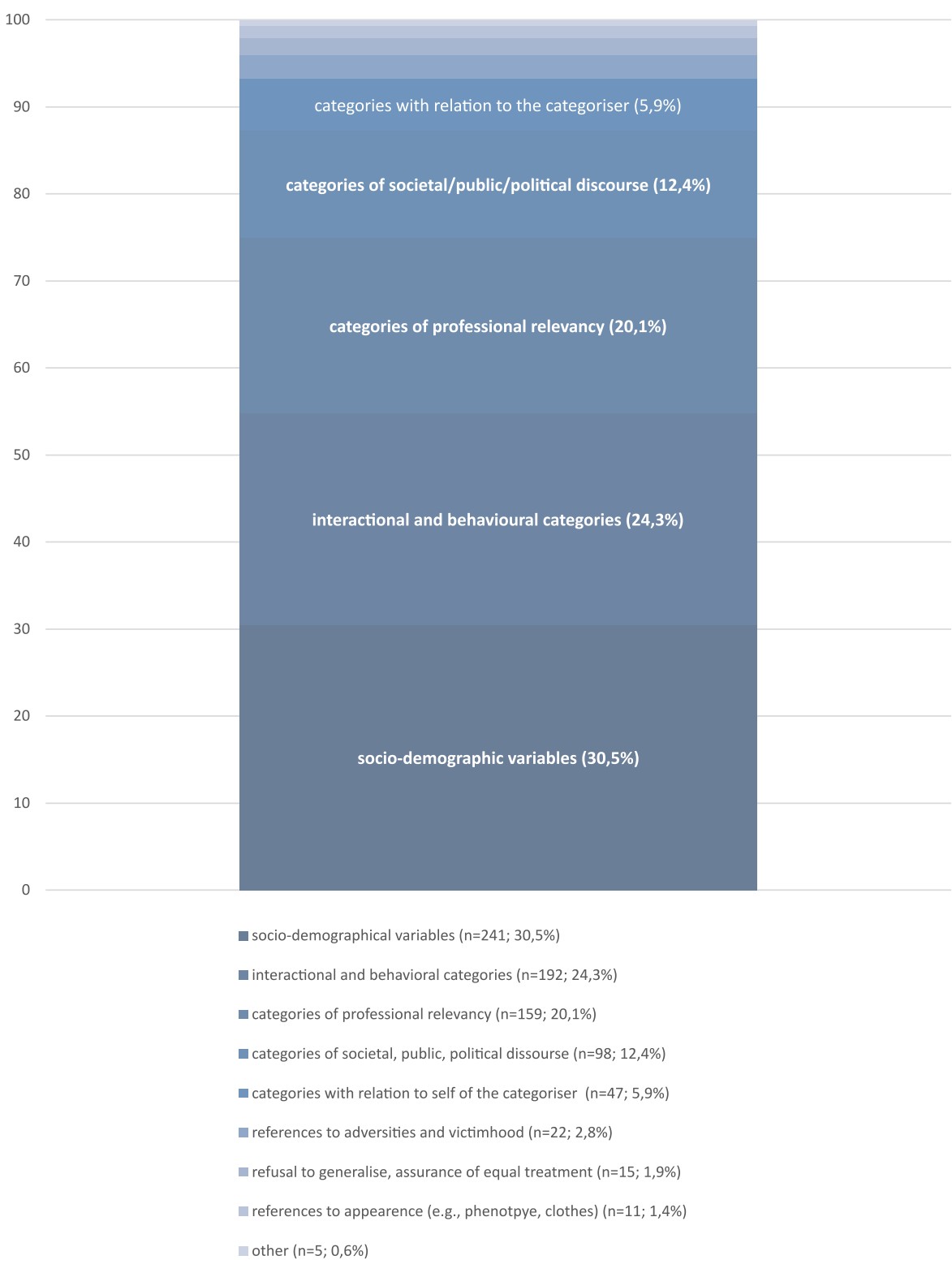

**Fig 3. Analysis of focus categories (inclusive duplicates, for frequencies < 8 see S1 Fig).**

"Blacks/Whites"

(L2D)

"Education (having more or less)"

(L1D)

"Politically prosecuted [ones]/economic refugees [. . .]

(S3E)

"Grateful-ungrateful [ones], e.g., I got a rose yesterday, but I have also been spat on once"

(L1D)

"Young Nations2 vs. old married couple from Nations2"

(A5D)

"Clothes: neat or scruffy"

(L1D)

"Differing cultural backgrounds: From an Nation7, an academic, with whom I can talk immediately: culturally and educationally similar, to a Nation1, who comes here without underwear in her robe, archaic, which for me is something quite foreign"

(A2D)

"Those who really have something [a medical condition] (should stay, I think e.g., a clever boy, dear person, then has bad luck, prostate cancer)/Those who rather have nothing (they are then not believed)"

(D3D)

"There are safe countries of origin and those with a prospect of getting a protected status"

(R1D)

Secondly–like described before–many social images are combined conglomerates of items, oftentimes containing socio-demographic variables, above all nationalities or regions of origin. For example, flight motives and prospects of staying are often attributed to specific countries or regions:

"War refugees from Nation19, Nation20."

(A4D)

"Nation12 and Nation4 come because children have deformities or chronic diseases [and have the hope that] modern medicine can do something about it."

(A1D)

"Nation2, largely labour migrants, fleeing from live circumstances, without being politically prosecuted."

(A1D)

"Economic refugees (plusnation2 [. . .]) you can see if they only come for money, I know Plusnations4 –I was told once, that there they have little money, [. . .] they want to join the

social system here, even if they have no chance [of staying] here (Plusnation29, Plusnation3, Plusnations4).”

(S2D)

“Nation12 [. . .], re-entered, they will never get a recognition, but they try”

(A1D)

We also observed that attitudes and “typical” behaviour in many cases get assigned regionally. People with specific origin were considered as being friendly, polite, open, impatient, aggressive/violent, hot-tempered, religious, demanding, calm, nice or more likely addicted.

“Drug addicts, young men from Plusnations22, addiction and Plusnations22 goes hand in hand”

(A4D)

“My favourite people are the Nation3s, they are nice and sweet”

(L3D)

“Nation6/Nation1, are calm, not aggressive, say please, pray–not like Nations4, who want to achieve everything through aggressiveness and violence.”

(S5D)

“Nation1 women—extremely self-confident, demanding, loud, aggressive, quickly forgotten then friendly again” (T3E)

Occasionally educational status and integration chances are also associated with coming from a certain country of origin. Only at two occasions attributions were explicitly assigned to specific ethnicities. But we observed–what we would call–an ethnitisation of nationalities, when German professionals used the term “Volk” (people/folk/lot) referring to specific nationalities, for example stating: “Nation3. They are a cheeky people, accept nothing” (F1D) or “My favourite people are the Nation3s [. . .]” (L3D) or talk about “Ethnic groups that make an effort to integrate, do various German courses at school, e.g., Nation5s make a lot of effort” (S3E). We encountered one biologising or racialising [120] statement: “Nation 1, used to be nomadic people: are tough, it´s still in their genes” (L1E).

Age and gender also emerge as meaningful social categories, that are associated with ways of being and behaving. More complex statements contain certain clusters of categorises–condensed into common prototypes–for example made up of gender, relationship status, region and/or religion. Here, for example, the female victim of sexual violence or human trafficking, from a certain region, travelling alone, sometimes in connection with unwanted pregnancy, is described. Or the young single man who is looking for a better life in Germany. Also, the religiously conditioned treatment of women is thematised together with nationalities, in two listed statements also referring to domestic violence.

Some categorisations of asylum seekers show a heightened awareness for the gender category. We observed professionals reflecting on own and foreign gender related attitudes and behaviours, frequently in combination with cultural ascriptions. Four female welfare professionals for example reflected on interactions with men from certain countries. One voiced surprise, since–in contrast to own prejudices–she felt treated very courteously and respectfully (T3E). Her colleague was worried, she might not be seen as trustworthy (T2E), another felt

one had to assert oneself (T1E) or should pay attention to clothing in a different way than usual (T4E).

Some professionals profess that they do not want to categorise (seven participants) and/or are treating everyone equally (five participants–four of them different ones), but no person has refused the exercise or has not continued listing after this statement, "I don't want to put people into boxes, but there are this and that kind of asylum seekers" was how the listing was started or continued in these cases. Professionals who problematised stereotyping, provided–on average–a little shorter list than others (13.6 compared to the general average of 16 mentioned items), professionals who assured the interviewer, they would treat everyone the same had on average longer lists of categorisations (M = 22).

What the consequences of social categorisation for interactions, decisions and ultimately care provision are, is not foreseeable from the exercise; asking about explicit categorisation does not automatically lead to further reflection on its consequences. Only in rare cases, we find meta reflections within individual statements:

> "How I categorise people is also influenced by political ideas and prior knowledge: The probability that people come for real asylum reasons influences how seriously I take them: Nations22 less, Nations1 are demanding and aggressive, you notice that. Then I went on holiday to Nation30, also to Nation4: those who are dumb are not there"
>
> (A2D)

However, the available data does provide us with the opportunity to look at continua of evaluations within social categorisation: We can identify forms of rather "neutrally" presented, condensed experiential knowledge as well as categorising statements that can be considered forms of Othering [121], where the differentiation contains depictions of radical distinction or inferiority [122]. We can check if deviations from the norm or own ideas of normality are pointed out [123], if pejorative terms are used, or negatively valued attributes are ascribed. Even when politically hotly debated, inherently negatively connoted categories like "economic refugees" were thematised, they could be presented as (seemingly) neutral observations: "The percentage of economic refugees is high" (S2D) or even substituted by empathetic descriptions pointing to people looking for better opportunities: "Those who want a better life" (S2E) or they could be accompanied by explicitly negative attributions, for example when a categoriser–using pejorative terms as main descriptor–insinuates that there were (predominantly) asylum seekers who want to exploit the German system:

> "[There are] normal refugees (100 out of 500) want to live here, go to school, work. Lazy/asocial [ones], come for the money: You are stupid if you work in Germany."
>
> (S3D)

Similarly, when thematising health seeking migration, it makes a difference if someone talks about a lack of resources for medical treatment in certain countries, the hopes of people to receive live-saving or altering treatment or speaks about asylum seekers as "medical tourists" (e.g., L1D). The health seeking migration category was in 9 of 15 cases presented with a negative connotation.

Such continua of–seemingly–neutral categorisations towards a heightened evaluative load of categories could be found throughout our data (see Table 6 for further examples):

**Table 6. Further example quotes showing an evaluative continuum of social categorisation.**

| Descriptive social categorisation | Evaluative social categorisation |
|---|---|
| "There are those, who need certificates" (A2D) | "80% simulants—want certificate ([because] have overslept the BAMF appointment or transfer)" (L1D) |
| "Grateful ones, there are many" (L2D) | "The best people I have seen are: Nation7, Nation31, you can never have a problem with a Nation31, they are grateful, [. . .] but Nation12: very aggressive "too bad, that's the truth, I've never seen a normal person there" (S5D). |
| "There are highly vulnerable mothers and children" (A3D) | "Nations5 are civilized people, they take care of their kids, Nations2 let them run free, Nations4 kids half and half" (S3D) |

## Discussion

### Summary and discussion of findings

The two most salient and therefore most relevant social categories that professionals in refugee settings referred to where "*demanding and expectant*" or "*polite and friendly*" behaviour of asylum seekers. Thirdly, the (presumed) effort that a person makes to *adapt* to the German environment, for example by learning the language was assessed by the professionals and used in clustering people. As fourth most important social category professionals mentioned "*economic refugees*".

In general, professionals seem to cluster the people in their care mostly according to their attitudes, which they derive from their behaviour in interactions with them. Both was frequently presented in combination with socio-demographic aspects, all ahead countries of origin, but also regions of origin and gender. Our data also show that social categorisation processes are permeable to public and political discourses, when (accepted and rejected) motives for flight or integration efforts and chances were thematised. Other common public topics like gender relations or cultural distance could be identified to a relatively small extend. An explorative comparison shows, that while health professionals often referred to health seeking migration, security contemplated maladaptation and insubordination and other staff often categorised polite and friendly behaviour.

With a focus on constructions and performances of deservingness, we want to discuss the two categories "economic refugee" and "demanding and expectant asylum seekers" in depth. We will also share some reflections on categorisation processes in the refugee context in general and the politicisation of professional action in this setting.

When professionals in reception centres decide if and how they are going to help the people that turn to them, they do not only consider entitlements and professional assessments, but also assess *whose* concerns "deserve attention, investment or care" [92, p. 2] or in other words "whose bodies, lives, and life chances matters [sic]" [91, p. 95]: Who *should* have access to a service or receive support?

Certain ideas about what proper help seeking behaviour looks like, play a role in ascriptions of deservingness (see "attitudes" and "reciprocity" in deservingness frameworks). Professionals appreciate patience, politeness and gratefulness, pressing and demanding behaviour is not welcome [124–126]. After interviewing administrative, social work and health care professionals within their ethnographic study Behrensen et al. [126] note:

> "While some asylum seekers succeed in qualifying for support in the eyes of the staff, others do not. The differentiation criterion that was mentioned again and again in the enquiry is the division into those who demand and those who need. It could also be described as "loud" and "quiet". Professionals prefer to assist the quiet ones rather than the loud ones"

(p. 98, own translation)

Altreiter–assessing social assistance work in Austria–similarly reports that demanding behaviour had a negative impact on ascriptions of deservingness [124]. Huschke [125] reviewed related literature and confirms that "docile, passive and shameful clients receive preferential treatment compared to demanding ones" [p. 352].

If certain behaviour seems to be expected to ascribe deservingness [91, 127], what is considered "proper" behaviour? Are expectations of how concerns and afflictions have to be brought before professionals culturally shaped? According to a psychological study on the specificity of virtues modesty and moderation were attributed a medium importance by German participants, "respect" however was considered highly important [128]. Demanding behaviour might be perceived as disrespectful by aid-providers and is therefore depreciated. In a German migrant health organisation, Huschke observed that "demanding too much and in a way that is perceived disrespectful" impacts professional decisions to the disadvantage of patients [p. 356]. Another interpretation could be, that certain ideas of justice might imply that people should not be favoured because they "shout the loudest" [124, p. 136f].

To be considered a deserving patient ore client, it might not be enough to be patient and quiet, you might be expected to show gratitude. Generally, the relationship between professionals and asylum seekers is asymmetrical, not only regarding resource distribution power but also relationally since it is not reciprocal. The patient or client does receive something but cannot offer an equivalent service in return [129], (see "reciprocity") which brings him/her in an inferior, "status-reducing position of mere gratitude" (ibid., p. 39). Especially if professionals emotionally expect or demand performances of gratitude as a counter-gift, this amounts to demanding an inner bow or subordination [129]. Persons behaving differently than expected, are perceived as resistant and oppositional and "run the risk of forfeiting their 'credit of compassion'" [130] with the professionals, their value is at stake because they are perceived as ungrateful and rebellious [130, 131] (see deservingness criteria "identity"). Our study identifies social categorisations according to the level of gratitude as well as evaluations of the level of adaption or subordination, which could point to such mechanisms. Unfortunately, a high level of personal commitment is often necessary for professionals to provide adequate care for asylum seekers. Caregivers frequently encounter structural and bureaucratic barriers in their efforts to uphold their professional values. They carry out their work "despite everything," against all odds, reacting to need and suffering. In return, some expect "nothing more than a little gratitude". If that is not granted (by the patient) they might feel betrayed in their efforts to help. Deservingness reasonings are "but loosely tethered to empirical realities and often carr [y] a powerful emotional charge" [91, p. 97]. That may explain, why we found *demanding-and-expectant* asylum seekers to be a salient social category. It does not mean there are many "demanding" asylum seekers, who expect too much of the professionals who are paid to be responsible for their care, but it might mean that refugee patients and clients who do not show the expected "proper" behaviour, leave such a lasting and negatively valued impression that an own social category has emerged.

Not only refugees, but patients and clients in general may be categorised, among other factors, based on their behaviour and the impact it has on professionals [132–136]. However, we believe the application of the discussed categories to refugees in reception settings has or can take on a different significance. 1.) The expectation to comply to a certain image of vulnerability, points to a powerful discourse, where asylum seekers are supposed to perform their deservingness, as "humble and grateful sufferers", not as "empowered subjects" who feel and enact a sense of entitlement [125, p. 353]. 2.) It is furthermore possible, that asylum seekers 'unfavourable' behaviour is less tolerated than if it was a person from the majority population (Kootstra

2016 found in an experiment studying public deservingness attitudes, that ethnic minority claimants were punished more severely for unfavourable behaviour [137]). Asylum seekers might be perceived as outgroup members, who not yet contributed to the solidarity community and therefore are–in social perception–not entitled to demand anything from it (see deservingness-criterion "reciprocity"). As mentioned, 3.) asylum seekers often depend on a high level of dedication of professionals to receive adequate care, if they lose the goodwill and commitment of professionals, it may have more severe consequences as it would have for a patient or client of the general population. This points to 4.) a greater power imbalance between refugee patients/clients and professionals, as–among else–asylum seekers can not as easily access care outside of the reception facility compared to a patient or client of the majority population who can easily change their doctor if they are dissatisfied or demand more or different care. This fact may contribute to the perception of being demanding, as the inhabitant may need to repeatedly ask the same professionals for assistance if their needs are not initially met. However, it is important to consider multiple factors contributing to specific social perceptions and classifications, some have been discussed.

Not only individual behaviour of asylum seekers impacts on their social image. Our results show, that also societally and politically imbued categorisations can be identified. Among else, the legitimacy of refugees to be in a country other than the one they fled from is thematised. Although the legitimacy of an asylum claim is not decided by refugee care professionals, but by policymakers and immigration officials, we found that the staff of outpatient-clinics and accommodation centres also differentiates asylum seekers according to flight motives and–in many cases–evaluates the legitimacy of those motives. Differentiations were made between constructed collectives of people who seem to have left their country apparently voluntarily contrasted with those who were forced to do so. According to the deservingness-criteria "control", their current need for assistance could than be framed as self-inflicted. This mirrors a political discourse, which already started in the 1980s, considering those as "good" refugees, who were politically persecuted or whose country was at war and others, often called "economic refugees", as "bad" ones, which were denied both economic usefulness and legitimate reasons for fleeing [66] (for a problematisation of this distinction of flight motives see Apostolova [48], who states, that it is based on a capitalist, neoliberal ideology fostering the illusion of economic relations being force-free, so that poverty and unemployment appear as free of (political) violence). Many studies have detected this social distinction in host societies [47, 51, 55, 57, 64, 65, 138–145] and related "tropes circulating the political and public discourse [. . .] where people seeking protection or better life opportunities are routinely framed as suspected criminals, 'tricksters' and potential welfare abusers" [146, p. 220]. In 1990 the term "safe countries of origin" became part of German asylum law, thus an assumption of illegitimacy on the basis of nationalities was legally established. After the peak in refugee inflow in 2015, yet another new term was politically introduced, now there is talk of "good" and "bad" prospects of staying [147, 148]. Especially people from so-called "safe countries of origin" and from countries with a past protection quota of lower than 50% [149] are assumed to have a poor prospect for a positive asylum decision (For a discussion of the lack of legitimacy of this pre-supposition, see [57, 148, 150]).

Our participants contemplate both, the presumed safety of certain countries and prospects of staying, mainly referring to bad prospects which they associate with certain countries. In their mind, this is oftentimes related to the planning of counselling and care processes (e.g., when considering, that a person may be deported before he or she can benefit from a certain service like psychotherapy or surgery), but it still might negatively impact decisions to the disadvantage of patients and clients. We find it highly likely that the thematisation of prospects of staying in refugee care settings–and public discourse–is just the "old" distinction between

"good" and "bad" refugees or wanted and unwanted refugees in a new guise. One that makes the construction of deserving and undeserving categories appear as being based on rational grounds [150].

„Categories are never just neutral descriptors of the world, used to report objectively on some state of affairs" [52, p. 167], like what the actual motivations of people are, to leave a certain country or what their actual prospects of a positive asylum decision are. "Rather, the act of categorising people into groups can work to accomplish particular tasks" [ibid.], "it orients to practical action" [151, p. 244]. This means, we need to ask: What social action is being accomplished in this particular instance? Describing, judging or making claims about others "reflect [s] and compose[s] moral reality" [152, p. 322] in inter-group relations [153]. Categories "may be deployed to make a social comment on asylum seekers' moral status and to present them as legitimate or illegitimate, deserving or undeserving, and welcome or unwelcome in this country" [52, p. 168]. Apostolova detects a consensus among the European member states, according to which "economic migrants" need to be kept out [48]. Constructing them as "undeserving Other" helps to justify repressive border controls [146, p. 1029].

At our field sites professionals are entrusted with caring for and protecting asylum seekers during their daily lives in Germany. Why would they have to distinguish them according to flight motives? What does it accomplish? As border and migration control cannot efficiently "keep" unwanted migrants "out" and the nation state cannot afford to openly cast their complete exclusion from social protection measures into laws, since neither international bodies nor whole societies approve the exclusion of non-citizens, so it seems the *internal* borderwork is implicitly delegated to non-state actors, including health care and other professionals [146, 154–156], that are involved in resource distribution decisions regarding people seeking international protection. They now also seem to screen for asylum eligibility criteria and denominate potentially undeserving recipients of *resources*, sorting people into "undeserving trespasser[s]" versus "those who deserve rights and care from the state" [127, p. 13].

The vague legal framework governing health and other care for newly arrived asylum seekers allows or even invites such social screening. It has been kept vague either out of overextension or as instrument of exclusion. Leaving ample room for discretionary decisions might be a result of the political impossibility to meet all opposing societal demands or it might make use of the assumption, that actors confronted with uncertainty, might be more restrictive in their discretionary decisions than if there were clear inclusive guidelines. In any case, the vague legal framework *allows* for political and social categorisations to influence prioritisation decisions of healthcare and other professionals. Refugee care might purposefully happen against the backdrop of ambiguity and suspicion. A "lack of explicit norms and procedures [. . .] blurs the boundary between necessary (and thus legitimate) professional discretion and discriminatory practice, both of which can be part of individual gatekeepers' trying to reconcile the politically unresolved conflict between health care and border care." [155, p. 69]. Distinguishing and evaluating asylum seekers flight motives and prospects of staying is of relevance to bordering and border control and less to decision-making of medical and other staff of reception centres. However, social categorisations in that setting show, that professionals may run the risk of being co-opted for border work when distributing resources.

The margins of discretion of all involved actors are navigated within a "culture of disbelief" [157] or a climate of suspicion [146], which affects the social perception of decision-makers. "Suspicion against racialised, mobile poor people circulates between the political sphere, public debate, and law and institutional practice" write Borelli et al. [146, p. 1026]. We observed social categories, pointing to suspicion in the daily communications of administrative, health, translation, social and security professionals. Our data show, that they ask themselves: Is this an economic or political refugee? Is it a "genuine" or "bogus" asylum seeker? Will this person

possibly be granted a protective status, or will he/she be deported? Does he/she really need help or only look for advantages and wants to exploit the system? Is this person really sick or faking illness just to get a certificate or create a reason to stay? Is it a real refugee or someone who only comes for better health care? "Suspicion is a systemic part of how law is implemented by street-level bureaucrats and plays a crucial role in their everyday decision-making–and, hence, becomes an institutionalised practice" [146, p. 1032, for references to mistrust, see 158]. The mere fact that moral negotiations are reflected in social categorisation processes, cannot say much about resulting actions and decisions, but we can assume that the state of suspicion [146] reflected in categorisations, affects interactions and relationships between asylum seekers and professionals.

Social categorisation happens quickly and often without conscious action, especially in the face of high arousal and stressful working conditions where quick decisions are necessary. Asked to explicate their social cognitions of asylum seekers, many participants problematised pigeonholing and shared their intention to treat everyone equally. Prejudice and stereotyping are not socially desirable, and the "majority of health professionals would find [. . .] [it] morally abhorrent, [but] they may not recognize manifestations of [. . .] prejudice and stereotyping in their own behaviour" as noted by Agyemang et al. (2007) [159, p. 241] (also see [160]). Neuro-scientific findings indicate that categorisation happens fast, is context dependent and contains different information simultaneously (like prior knowledge, motivation, social expectations, perception of facial expressions) [161]. Behaviour is not necessarily always controlled by conscious cognitive processes, since the responsible reflective system provides a relatively slow, rule-based processing of information; if motivation [162] or energy [163] is low or the capacity limited "less elaborate processing takes place." [162, p. 41f] Capacity can be limited, if circumstances are suboptimal, e.g., time pressure or high arousal [ibid.]. Then behaviour is rather controlled via the impulsive system, which works associative, fast, and efficient (ibid.). A high workload and strenuous circumstances often characterise the field of refugee care, we also already discussed a high emotional load regarding moral negotiations of deservingness. This work environment therefore facilitates quick social categorisations and judgements.

## Methodological limitations, strength, and implications for future studies

This exploratory study can only provide insights into real-world social categorisation of professionals at specific times in two specific places. Public discourses, deservingness negotiations, and social categorisation practices are dynamic and may change over time, they may also be distinct between locations. Thus, the findings, while similar observations could be made elsewhere in Germany, should not be generalised.

In a qualitative chapter, we show the differences in the "quality" of the categorisations, since the semi-quantitative free listing analysis can not grasp the complex spectrum and sometimes intersection of evaluations. The need to align wordings and simplify more complex statements by breaking them up into singular units of meaning might be the biggest limitation of our study. It should and has been done intersubjectively in groups of researchers, since for the alignment and qualitative coding of items and especially the assignment of codes to super- and focus codes, discussion is needed and recommended.

A further interpretive difficulty lies in the interpretation of the salience index, since there is no standardised salience threshold, so it is "a matter of judgement" [164, p. 1438] to determine how many social categorisations can be considered as most relevant in this context. Also, the decision to adapt variables pointing to origin of asylum seekers for a second analysis round can be criticised, because we *only* took national and regional specifications out and other socio-demographic variables like mentioned age groups and genders were still counted

separately. This was decided since references to the specific origin of asylum seekers were most frequently mentioned, but the mentioned specific nationalities and regions did not carry much analytical value in their pseudonymised form (also see explanation in the methods chapter).

Furthermore, in a conventional free-list analysis duplicates are deleted since only the position at which an item is *firstly* mentioned matters for the index calculation. We assumed mentioning categories more than once also might say something about their importance or indicate, that a social perception has particularly preoccupied participants, so we did all additional analysis steps including the duplicates. The decision to analyse super and focus categories based on this original data could be debated, since it may complicate understanding and comparability with the salience calculation results.

Some further limitations we see in the prompt, the sample size, and the placement of the exercise in the research context. We used the prompt: "What kind of asylum seekers are there?" since we aimed for an identical prompt for professionals from different disciplines. This rather "official" terminology might have triggered more associations that are drawn from the public discourse, than if we would have used individual professional descriptors like: patients, clients, or inhabitants. However, Augoustinos and Quinn [165] found that different social categories (in their case: illegal immigrants, asylum seekers, refugees) did not elicit different trait attributions to these categories, only attitudinal judgements differed. Many of our participants replaced our prompt in their answer with "refugees", so it might have been advisable to have used this term in the prompt.

Our sample size is sufficient for a general free-list analysis [166], but too small for a meaningful comparative analysis of categorisations of the different professions. Nonetheless, to get an impression of most frequently used categories by different professionals, we conducted an additional analysis of three subgroups. It is important to approach these findings with caution, due to the even smaller sample sizes. Particularly in the super- and focus category analysis, this is problematic since proportions are distorted by multiple mentions of items by individual participants, therefore results are only provided as Supporting Information. The overall number of participants was pre-determined and therefore limited by the embeddedness of the exercise in an ethnographic case study at two field sites. In this embedding lies a further limitation, since the free-listing happened at different times during the study and with different relational background of the researcher to the study subjects, meaning some lists were generated directly after in-depth interviews or after sharing daily work experiences, others were generated–literally–in the hallway, without much prior communication between researcher and participant, this might have influenced the results in different ways.

Because of our research focus on health equity issues, we discussed some examples of categorisations, that *could* be of concern for the health sector. Patient categories have been found to influence physician behaviour [3] and evidence indicates that biases are likely to influence diagnosis and treatment decisions [2, 167] as well as interactions and health outcomes [26]. However, that deservingness is negotiated by professionals does not mean, that asylum seekers that are judged as undeserving will not be provided with proper care, it just means that deservingness is negotiated. It can be negotiated on the basis of social categories, that asylum seekers got assigned to. Categorisation itself *does not imply* stigmatisation, discrimination, or oppression [108]. Other studies are needed, to systematically assess a possible link.

Taking civic stratification [168], superdiversity [43, 44, 169] and intersectionality [170] seriously means to firstly explore real-world human differentiations, instead of only working with pre-conceived categories, before studying professional decision-making, biases, Othering, and potential discrimination. Free-listing presents itself as a useful, exploratory, quickly applied method to do so. It has already gained some popularity in the public health context [166, 171–

173] and can be fruitfully applied in mixed-methods projects in the fields of equity studies. It seems advisable to use it in combination with other field approaches, so that the enquiry can take the complexity of social spaces and discourses into account. Starting with exploring categorisations of the study subjects helps researchers to emancipate from "policy categories" [174, 175] and continuously account for the dynamic nature of social attribution processes in modern, plural societies in their research.

## Conclusion

We explored real-world social differentiations of asylum-seeking patients, clients and inhabitants by German professionals working in reception centres. We also could identify themes associated with their categorisations. We found that behaviours and attitudes, as well as socio-demographic variables such as nationality of origin, play a significant role in the social categorisation of asylum seekers. Focussing on two identified social categories: "demanding and expectant asylum seekers" and "economic refugees", we discussed negotiations and justifications of deservingness in this context. Our findings highlight that social distinctions that organisations and their professionals make, are permeated by political and societal discourses regarding the individuals under their care.

To be considered as deserving, following our results and discussion, an asylum seeker apparently has to show the proper behaviour, be perceived as being in "real need" (however defined) and must be the "proper kind" of refugee. We assume that such considerations negatively affect, at least, the relationship and interactions between asylum seekers and the professionals they depend on for what they need while being accommodated in reception centres.

If explicitly or implicitly, consciously, or unconsciously professionals who take care of asylum seekers enact politics and become political actors themselves, while they navigate their margins of discretion and negotiate deservingness [176, 177]. Refugee care takes place against the backdrop of the highly contested field of immigration policy, maybe we can even go so far as to say, that therefore every encounter between professionals and asylum seekers can be interpreted as being simultaneously a political act, or at least having a political dimension. The professionals cannot escape this, they must, for example, behave in an appreciative, accepting way towards restrictive guidelines or resist them openly or covertly. The same goes for social classifications of refugees that permeate their society–and thus also themselves: one can reflect on them or not–in which case they might be no less effective–we can approve or disapprove of our own or others´ classifications. In this politicised field of action [178] there might be no neutral position and therefore positioning. So, if politics will anyways be relevant, should organisations take a clear stand and formulate an explicit political mission instead of believing or pretending to only follow professional rationales? Organisational culture development in this direction could start by exploring and consciously unpacking social categorisation practices of their members.

## Supporting information

**S1 File. Codes of the free-list analysis and assigned super-categories.**
(PDF)

**S2 File. Flame analysis—Frequencies (inclusive nation/region), super- and focus category analysis of professional groups.**
(PDF)

**S1 Table. Descriptives of two rounds of the free-list analysis in flame.**
(PDF)

**S2 Table. Top 10 absolute frequencies.** Incl. duplicates, with/without nation/region specification.
(PDF)

**S1 Fig. Bar chart of super-categories (incl. descriptions of frequencies < 8).**
(EPS)

## Acknowledgments

We thank Peter Geisler and Johannes Wischert for their technical support. PG for his help with salience calculations in Excel and JW for his help with figure formatting.

## Author Contributions

**Conceptualization:** Sandra Ziegler.

**Formal analysis:** Sandra Ziegler.

**Funding acquisition:** Kayvan Bozorgmehr.

**Investigation:** Sandra Ziegler.

**Methodology:** Sandra Ziegler.

**Project administration:** Kayvan Bozorgmehr.

**Resources:** Kayvan Bozorgmehr.

**Supervision:** Kayvan Bozorgmehr.

**Validation:** Kayvan Bozorgmehr.

**Visualization:** Sandra Ziegler.

**Writing – original draft:** Sandra Ziegler.

**Writing – review & editing:** Kayvan Bozorgmehr.

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
