## [Decision Letter · Decision Letter 0]

2 Oct 2023

PGPH-D-23-01416

“I don´t put people into boxes, but…” A free-listing exercise exploring social categorization of asylum seekers through health professionals and staff in two German reception centers

Dear Dr. Ziegler,

Thank you for submitting your manuscript to PLOS Global Public Health. After careful consideration, we feel that it has merit but does not fully meet PLOS Global Public Health’s publication criteria as it currently stands. Therefore, we invite you to submit a revised version of the manuscript that addresses the points raised during the review process.

Please consider the comments and suggestions provided by the two reviewers when revising your manuscript.

In particular, please make sure to introduce the concept of deservingness, to elaborate on the interactions between public discourse and social categorization, to expand on the methods used, to critically reflect on the link of the findings with migration status (or lack thereof), and to align the conclusions with the study goals.

We look forward to receiving your revised manuscript.

Kind regards,

Nora Gottlieb

Academic Editor

Journal Requirements:

Additional Editor Comments (if provided):

Reviewers' comments:

Reviewer's Responses to Questions

**Comments to the Author**

1. Does this manuscript meet PLOS Global Public Health’s publication criteria? Is the manuscript technically sound, and do the data support the conclusions? The manuscript must describe methodologically and ethically rigorous research with conclusions that are appropriately drawn based on the data presented.

Reviewer #1: Yes

Reviewer #2: Yes

2. Has the statistical analysis been performed appropriately and rigorously?

Reviewer #1: N/A

Reviewer #2: I don't know

3. Have the authors made all data underlying the findings in their manuscript fully available (please refer to the Data Availability Statement at the start of the manuscript PDF file)?

Reviewer #1: Yes

Reviewer #2: Yes

4. Is the manuscript presented in an intelligible fashion and written in standard English?

Reviewer #1: Yes

Reviewer #2: Yes

5. Review Comments to the Author

Reviewer #1: Dear authors,

thank you very much for this intriguing contribution to the growing body of literature on social categorisations of (forced) migrants. While most contributions to this research field in the social sciences are of a conceptual or theoretical nature, the present article offers an attempt to empirically grasp the way how asylum seekers/refugees are categorised, how those social categorisation processes can be understood in light of ongoing policy and public debates, and what consequences those categorisations might have with regards to the perpetuation of public discourses about forced migration.

Given the very scarce empirical research into categorisation processes of (forced) migrants, the explorative approach of free-listing is an appropriate way of generating knowledge about the way how forced migrants are perceived by different groups of professionals. The mixed-methods design is described in an appropriate way that also shows that the authors are aware of the limitations to this approach.

The text is well-written in plain English. The conclusions drawn from the results are generally plausible, although some further explanations might be needed, as I will account for in more detail below.

In general, I recommend publication of this article, which offers insights that might lead to a better understanding of the relevance of social categorisation processes in the context of forced migration. However, I would like to offer some suggestions to the authors for consideration:

1) The results show very distinct ways of categorising forced migrants. Some are related to their perceived behaviour (thankful, polite, aggressive, etc.), some relate to their status ("economic" refugees, legitimate refugees, bogus asylum seekers, etc.), and others relate to their national or ethnic background. Whereas the last two categories clearly refer to forced migrants, one might wonder if the first category refers to patients in healthcare or people in general. Maybe if asked "what kind of PEOPLE are there" the same answers from the first category could have emerged. Obviously, it is impossible to answer this question in retrospect, however it might be worth a reflection in the paper. These freelistings might imply that the professionals think about asylum seekers / refugees in those categories, it might also be that - particularly medical practitioners - in general think about patients in these categories regardless of their migration or non-migration status. The latter would also be an interesting finding but leads to different conclusions (maybe the respondents who listed those categories do not really distinguish between asylum seekers / refugees and non-mobile populations).

2) Given the distinct groups of professionals that were included in the free listing exercise and the potentially different outcomes, I am not sure if I agree with the authors that a disctinction of the results by groups should be avoided. In order to have higher numbers of respondents in each group, groups could be merged, for instance into health personnel, support personnel and security personnel.

3) Although the concept "deservingness" seems appropriate in this context, it needs a bit more of an explanation about the way how it emerged in this context (from the data, from the literature, from public discourses), hence why is it used to embed the findings in the discussion part but not introduced in more detail, for instance in a theoretical background section or in a literature review section? It might help the reader to appreciate the reference to this concept more if its meaning for understanding the results were introduced in a bit more comprehensive way.

Reviewer #2: I have carefully reviewed your manuscript and would like to provide a constructive and respectful peer review. Overall, I appreciate the effort put into this research and the significance of the topic. Here are my comments on each segment of your paper:

Abstract: The abstract provides a concise overview of your study; however, it could benefit from a stronger connection with your manuscript. It is crucial that the abstract mirrors the research questions and findings presented in the manuscript to ensure coherence. Additionally, highlighting the positive results and their relevance to the research questions would make the abstract more impactful and of interest to readers.

Introduction: The introduction effectively sets the stage for your research by providing background information and linking it to the importance of social categorization in the context of asylum seekers and immigration. You successfully establish the relevance of social categorization with references to previous literature. To enhance your argument, consider elaborating on the interaction between political and societal discourse and its impact on social categorization, as this seems central to your study. Furthermore, clarifying the connection between your two research questions and exploring the potential implications of discrimination and public discourse on social categorization would strengthen your introduction. It would also enhance the introduction to provide background information on deservingness.

Materials and Methods: While I am not an expert in qualitative research, it is essential to ensure that your methods are clearly explained for the benefit of all readers. Consider providing additional details about data processing and analysis, particularly in the context of multiple-step qualitative analysis and the three-step content analysis. Addressing the issue of generalizability is important, and you might want to discuss the limitations of applying findings from a single social unit to a larger population of similar units. It's commendable that you mention the approval of your study by the university ethics committee.

Results: To improve the clarity of your results section, consider describing the variables presented in the tables to allow readers to interpret the tables independently. Additionally, it would be beneficial to keep the results section focused solely on presenting the results, without delving into the methodology used to obtain them, this should be mentioned in the methods section. Avoid providing interpretations in this section and present the results in the same order as your research questions were posed in the introduction.

Discussion: Start the discussion with a concise summary of your research's positive results and their alignment with the existing literature. To enhance clarity, use a consistent term to refer to social categorization throughout this section. Expanding on the relationship between social categorization and public discourse, in addition to the discussion of deservingness, would strengthen the overall argument. Frame the discussion in a way that ties your research findings to the existing literature. Your acknowledgment of study limitations is appreciated, but consider presenting these after discussing your findings and highlighting your study's strengths.

Conclusion: In your conclusion, focus on summarizing your research questions and the corresponding findings. Ensure that your conclusions align with the scope of your study and avoid introducing new elements.

In summary, your study presents valuable primary research, and the topic's relevance is clear. However, some improvements are needed in terms of clarity and organization in various sections of the paper. While I cannot confirm the novelty of your results, I encourage you to clarify and expand upon your methods, align your conclusions with your research questions, and address the points raised in this review. It is my belief that with these revisions, your manuscript has the potential for publication.

6. PLOS authors have the option to publish the peer review history of their article (what does this mean?). If published, this will include your full peer review and any attached files.

**Do you want your identity to be public for this peer review?** For information about this choice, including consent withdrawal, please see our Privacy Policy.

Reviewer #1: No

Reviewer #2: No

---

## [Editor Report · Decision Letter 1]

23 Jan 2024

“I don´t put people into boxes, but…” A free-listing exercise exploring social categorisation of asylum seekers by professionals in two German reception centers

PGPH-D-23-01416R1

Dear M.A. Ziegler,

We are pleased to inform you that your manuscript '“I don´t put people into boxes, but…” A free-listing exercise exploring social categorisation of asylum seekers by professionals in two German reception centers' has been provisionally accepted for publication in PLOS Global Public Health.

Best regards,

Julia Robinson

Executive Editor